# Influence of Mixed Valence on the Formation of Oxygen Vacancy in Cerium Oxides

**DOI:** 10.3390/ma12244041

**Published:** 2019-12-05

**Authors:** Gege Zhou, Wentong Geng, Lu Sun, Xue Wang, Wei Xiao, Jianwei Wang, Ligen Wang

**Affiliations:** 1General Research Institute for Nonferrous Metals, Materials Computation Center, GRIMAT Engineering Institute Co. Ltd., Beijing 100088, China; zgg8997@126.com (G.Z.); sunlu@grinm.com (L.S.); wangxue@grinm.com (X.W.); xiaowei@grinm.com (W.X.); wangjianwei@grinm.com (J.W.); 2School of Materials Science and Engineering, University of Science and Technology Beijing, Beijing 100083, China; geng@mater.ustb.edu.cn

**Keywords:** cerium oxide, electron valence state, first-principles calculation, oxygen vacancy

## Abstract

Ceria is one of the most important functional rare-earth oxides with wide industrial applications. Its amazing oxygen storage/release capacity is attributed to cerium’s flexible valence conversion between 4+ and 3+. However, there still exists some debate on whether the valence conversion is due to the Ce-4f electron localization-delocalization transition or the character of Ce–O covalent bonds. In this work, a mixed valence model was established and the formation energies of oxygen vacancies and electronic charges were obtained by density functional theory calculations. Our results show that the formation energy of oxygen vacancy is affected by the valence state of its neighboring Ce atom and two oxygen vacancies around a Ce^4+^ in CeO_2_ have a similar effect to a Ce^3+^. The electronic charge difference between Ce^3+^ and Ce^4+^ is only about 0.4*e*. Therefore, we argue that the valence conversion should be understood according to the adjustment of the ratio of covalent bond to ionic bond. We propose that the formation energy of oxygen vacancy be used as a descriptor to determine the valence state of Ce in cerium oxides.

## 1. Introduction

CeO_2_ is a widely used material in various important technological fields, such as solid oxide fuel cells [1] and three-way catalysts [2]. It is a remarkable feature that Ce can be flexibly converted between a valence of 4+ and a valence of 3+. This leads to the easy generation of oxygen vacancies in the crystal lattice and its high oxygen storage capacity. The Ce valence state change also has a direct and significant influence on its catalytic efficiency. It is still widely debated where the extra two electrons go after an oxygen vacancy is created. Some researchers argued that the two electrons left behind by the oxygen vacancy are localized to two neighboring Ce atoms and the corresponding Ce atoms are reduced to a valence state of 3+ [3]; others claimed that the extra two electrons are localized to four nearby Ce atoms that have a fractional valence state of 3.5+ [4]. There are also arguments about the electron occupancy of Ce-4f orbital in CeO_2_. While Wuilloud [5] believed that Ce-4f was almost unoccupied and that Ce ions presented a 4+ valence, Fujimori [6] and Normand et al. [7,8] believed that Ce-4f electrons should be in a 0.5 occupation state so that Ce presented a fractional or intermediate valence state more similar to covalent bonds. Koelling et al. [9] found charge overlaps between Ce and O and claimed that the bonding mode of Ce–O should be a covalent bond. Due to the covalent bonding character, the valence state for Ce must be in the transition zone of 3+ ~ 4+. In fact, Ce–O bonding was found to have a visible polarization that is not typical for a pure ionic bonding [10]. The quantum theory of atoms in molecules QTAIM analysis [11,12] showed that the electron density and its associated Laplacian could be partitioned into covalent, ionic, and resonance components [13]. The electron-pair bonding is usually a continuous spectrum of intermediate situations stretching between the covalent and ionic extremes [13]. Therefore, the argument about the Ce valence in the transition zone of 3+ ~ 4+, based on the covalent bonding character, is plausible. For the Ce valence state, the following four scenarios may exist: (1) Ce has a valence of 4+ in CeO_2_; (2) the Ce valence state is reduced within the range of 3+ ~ 4+ when few oxygen vacancies or n-type dopants exist in cerium oxides; (3) Ce cations have two valence states of 3+ and 4+, i.e., a mixed valence model, when there are multiple oxygen vacancies in a local area of CeO_2_ to completely reduce some Ce to the 3+ valence Ce; (4) in Ce_2_O_3_, the Ce atoms are a complete valence of 3+. When Ce is in the region of the 3+ ~ 4+ valence state, it is generally accepted that oxygen vacancy has an effect on the change of its valence state. But to what extent do oxygen vacancies affect the Ce valence state? If there is a pre-existing 3+ Ce, how will it affect the formation of oxygen vacancies? The answers to these questions are unclear. To date, how the Ce valence state responds to the continuous generation of oxygen vacancies remains largely unknown. 

We know that it is harder to form an oxygen vacancy in Ce_2_O_3_ than in CeO_2_ because in Ce_2_O_3_ the Ce atoms are in the reduced state. When a Ce atom changes its valence state from 4+ to 3+, making itself similar to Ce^3+^ in Ce_2_O_3_, it will become more difficult to form an oxygen vacancy around this Ce atom. On the other hand, if oxygen vacancies are continuously introduced around a 4+ Ce in CeO_2_, the formation of oxygen vacancy will also become more and more difficult. At a point, the formation energy of oxygen vacancy around a Ce^4+^ in CeO_2_ reaches the formation energy of oxygen vacancy in Ce_2_O_3_. At this point, we can consider the Ce atom to have completely changed its valence state from 4+ to 3+. In this paper, we propose to use the formation energy of oxygen vacancy as a descriptor or prober to detect the valence state of Ce in cerium oxides. In order to do so, we built a mixed valence model following Takenori Yamamoto et al. [14] and calculated the formation energy of oxygen vacancy in the mixed valence supercell. Although the formation energy of oxygen vacancy is larger when it is closer to Ce^3+^ than to Ce^4+^, the results show that they all are lower than that of Ce_2_O_3_. This may be understood according to the features of Ce–O covalent bonds and the adjustment of the ratio of covalent bond to ionic bond.

## 2. Computational Method 

The calculations were performed within the framework of density functional theory as implemented in the Vienna ab initio simulation package (VASP.5.4.4, University of Vienna, Wien, Austria) [15]. The electron–ion interaction was described using the projector augmented wave (PAW) method [16] with valence 5s^2^5p^6^5d^1^4f^1^6s^2^ for Ce and 2s^2^2p^4^ for the O atom and the electron exchange and correlation were treated within the generalized gradient approximation (GGA) in the Perdew–Burke–Ernzerhof (PBE) [17] form. The cut-off energy for the basis set was set to 500 eV for all the systems. In order to account for the strong on-site Coulomb repulsion among the Ce 4f electrons, a Hubbard parameter U was also included and U was set as 5 eV [18,19]. The supercells (see Appendix A) employed in our calculations are shown in Figure 1. The supercells employed for CeO_2_ and Ce_2_O_3_ calculations are relatively small but they have similar lateral dimensions to those in the mixed valence model. The 16 × 16 × 16, 16 × 16 × 8 and 16 × 16 × 4 Monkhorst-Pack k-meshes were used for the Brillouin zone sampling for CeO_2_, Ce_2_O_3_, and the mixed valence system, respectively. The atomic positions and cell parameters were allowed to relax in the calculations until the forces on all atoms were smaller than 0.01 eV/Å, and the total energies were converged to 10^−5^ eV. We performed non-spin polarized calculations for these systems. The optimized structure parameters are listed in Table 1, together with the experimental and other theoretical values [14,20,21,22,23,24,25] for comparison. It can be seen that the calculated lattice parameters of CeO_2_ and Ce_2_O_3_ are consistent with the experimental values. For the mixed valence model, we obtained a = 3.826 and c = 15.546, which are comparable to 3.822 and 15.351 by Yamamoto et al. [14].

## 3. Results and Discussion 

The formation energy of an oxygen vacancy is calculated by
(1)$$E_{f}^{V_{o}}=E^{\mathrm{tot}}\left(V_{o}\right)-E^{\mathrm{tot}}\left(\mathrm{ref}\right)+\frac{1}{2}E\left(O_{2}\right),$$
where $E^{\mathrm{tot}}\left(\mathrm{ref}\right)$ is the total energy of the reference system, and $E^{\mathrm{tot}}\left(V_{o}\right)$ is the total energy of the supercell that is established by removing one oxygen atom from the reference system. The reference system may have included oxygen vacancies. $E\left(O_{2}\right)$ is the total energy for the ground state of an optimized oxygen molecule in the gas phase. 

According to Equation (1), we obtain the formation energy of an oxygen vacancy in CeO_2_ to be + 4.48 eV, which is in agreement with the experiment values of + 4.68 [26] and 4.2 ± 0.3 eV [27]. The theoretical results obtained by DFT (Density Functional Theory) calculations vary in the large range of 2.84~6.74 eV [27]. There are two kinds of oxygen sites in cerium trioxide. The calculated oxygen vacancy formation energies are 5.4 eV and 5.7 eV, respectively. The lower oxygen vacancy formation energy of 5.4 eV was selected in the following discussions. Our calculated results are consistent with the non-reducible character of Ce_2_O_3_, requiring the large energy (> 5.0 eV) to form an oxygen vacancy [24]. In the mixed valence state model, there exist five possible inequivalent oxygen-vacancy sites, as shown in Figure 1. The calculated formation energies for these oxygen vacancies are given in Table 2 and Figure 2. Among the five possible sites, the oxygen vacancy $V_{O1}$ has the lowest formation energy of 4.79 eV. By comparing the formation energies of oxygen vacancy at five different sites in the mixed valence state, the formation energies of $V_{O4}$ and $V_{O5}$ are the most positive, reaching 5.32 eV, which is about 0.5 eV higher than that for $V_{O1}$. This is because there is a Ce^3+^ atom near sites 4 and 5, and it is more difficult to form an oxygen vacancy around a Ce^3+^ than around a Ce^4+^. In Table 2, the oxygen vacancy formation energies around a Ce^4+^ in CeO_2_ are also given. We can see that the oxygen vacancy formation becomes increasingly more difficult when oxygen vacancies around the Ce atom already exist. It is worth noting that the formation energies of the first two oxygen vacancies are smaller than the formation energy of an oxygen vacancy in Ce_2_O_3_ and those for $V_{O4}$ and $V_{O5}$ in the mixed valence state model. However, the formation energy of the third oxygen vacancy around a Ce^4+^ in CeO_2_ is larger than that in Ce_2_O_3_. In other words, it is more difficult to generate the third oxygen vacancy around a Ce atom in CeO_2_ than to generate an oxygen vacancy in Ce_2_O_3_. This means that the presence of the 3+ cerium has a larger effect than that of one oxygen vacancy and a similar effect of two oxygen vacancies around a Ce^4+^ in CeO_2_. We may also argue that it might not result in two Ce^3+^ when one oxygen vacancy is generated in CeO_2_. The Ce valence state is more likely to be between 3+ and 4+. This argument is consistent with the existence of covalent bonds in cerium dioxides. 

Bader’s charge analysis was used to obtain the total electronic charges of the atoms in CeO_2_, Ce_2_O_3_, and the mixed valence system. The results are presented in Table 3. We can see that in the mixed valence model, the oxygen and cerium atoms in the Ce_2_O_3_ region have higher electronic charges compared to those in the CeO_2_ region. For Ce3 and Ce4, they have about 0.4 more electrons than other Ce atoms in the mixed valence model. Except for Ce3 and Ce4, the other Ce atoms in the supercell have charges close to those Ce atoms in CeO_2_ bulk. This implies that Ce1, Ce2, and Ce5 have an environment similar to CeO_2_. We normally consider Ce to have a valence of 4+ in CeO_2_ and 3+ in Ce_2_O_3_. However, the Bader charge difference between 3+ Ce and 4+ Ce is only about 0.4*e*. Therefore, it is believed that the valence state adjustment is actually achieved by adjusting the ratio of covalent bond to ionic bond. This notion that cerium oxides have the features of covalent bonds is consistent with previous studies [9]. The charge density distribution of cerium dioxide is plotted in Figure 3. It shows a charge bridge between the Ce and oxygen centers [9]. Therefore, it was considered to form covalent bonds between the Ce and oxygen centers [9]. However, because the electronegativity difference between Ce and O is about 2.0 [28], the Ce–O bonds should include an ionic bond character. To quantitatively characterize covalent and ionic bonding, a QTAIM analysis [11,12,13] of the topology of the total density should be performed at the bond critical point.

The partial densities of states (PDOSs) for Ce^3+^ and Ce^4+^ in the mixed valence model and the Ce atom next to two oxygen vacancies in CeO_2_ are shown in Figure 4. From this, we can see the following: (1) the 4f state is unoccupied for Ce^4+^ in the mixed valence model, while it is partially occupied for Ce^3+^; (2) the O 2p-Ce 4f gap is larger for Ce^4+^ than that for Ce^3+^. Our previous study [29] indicated that the 4f state of Ce^3+^ in Ce_2_O_3_ splits into occupied (4f^1^) and unoccupied (4f^0^) states, rendering a Ce_2_O_3_ a Mott insulating phase. In the mixed valence model, while we do not restrict the electron to completely localize at the Ce atom, the partial occupation of the Ce^3+^-4f state may be because it is similar to a polaron in CeO_2_ [30]. Figure 4c shows that after introducing two oxygen vacancies around a Ce^4+^ in CeO_2_, its PDOS is very similar to that of Ce^3+^ in the mixed valence model, i.e., the 4f state is partially occupied and the O 2p-Ce 4f gap becomes narrower than that for Ce^4+^. The multiple peaks in the Ce 4f state may be attributed to the lattice distortion due to the oxygen vacancies. This also verifies, from another angle, that the valence state change of Ce is accompanied by generating oxygen vacancies around a Ce atom.

## 4. Conclusions

In this paper, the valence states in cerium oxides were investigated by performing density functional theory calculations based on a mixed valence model. The formation energies of oxygen vacancies and the electronic charges in the mixed valence model were calculated. We found that the formation energy of oxygen vacancy is affected by the valence state of its neighboring Ce atom, and propose that the formation energy of oxygen vacancy can be used as a descriptor to determine the valence state of Ce in cerium oxides. Our results show that two oxygen vacancies around a Ce^4+^ in CeO_2_ have a similar effect to a Ce^3+^. This is also supported by the partial densities of states that, for both cases, the 4f state is partially occupied and the O 2p-Ce 4f gap becomes narrower than that for Ce^4+^. The calculated results further show that the electronic charge difference between Ce^3+^ and Ce^4+^ is only about 0.4*e*, far from the nominal charge difference of 1.0*e*. Therefore, the valence state conversion should be understood according to the characters of Ce–O covalent bonds and the adjustment of the ratio of covalent bond to ionic bond.

## Figures and Tables

**Figure 1 materials-12-04041-f001:**
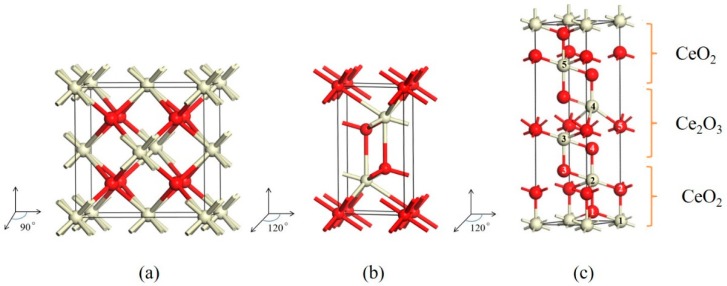
(**a**) Fluorite-type structure of CeO_2_. (**b**) Hexagonal A-type structure of Ce_2_O_3_. (**c**) CeO_2_/Ce_2_O_3_ mixed valence structure model. Ce atoms are represented by large grey spheres, and O atoms by small red spheres.

**Figure 2 materials-12-04041-f002:**
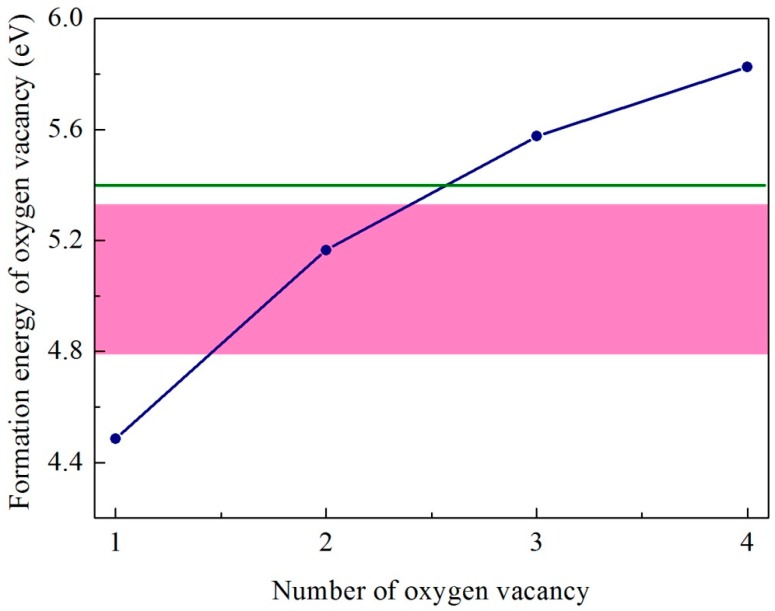
The calculated formation energy of oxygen vacancy as a function of the number of total oxygen vacancies around a Ce atom in CeO_2_. The shaded region represents the range of oxygen vacancy formation energy in the mixed valence structure. The horizontal line represents the formation energy of oxygen vacancy in Ce_2_O_3_.

**Figure 3 materials-12-04041-f003:**
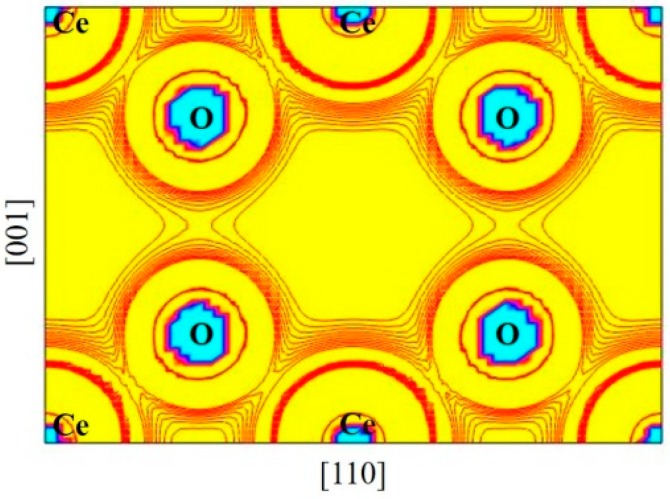
The valence charge density of the (110) plane for pure cerium dioxide.

**Figure 4 materials-12-04041-f004:**
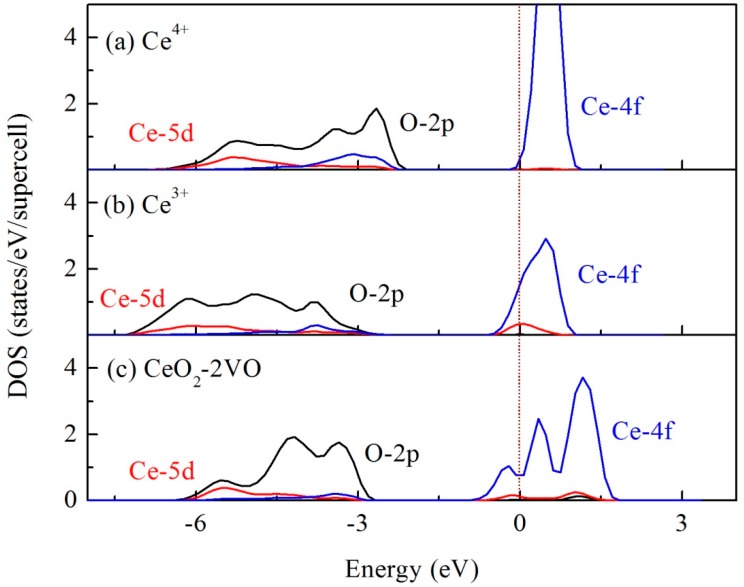
Partial densities of states (PDOS) state density distribution of mixed valence state model: the black, red, and blue lines represent the O-2p, Ce-5d, and 4f orbital electrons, respectively. The dotted line represents the Fermi level: (**a**) Ce^4+^ in mixed valence model (**b**) Ce^3+^ in mixed valence model (**c**) Ce nearest to two oxygen vacancies in pure ceria.

**Table 1 materials-12-04041-t001:** Our calculated equilibrium lattice parameters (in Å), together with the experimental and other theoretical values [14,20,21,22,23,24,25] listed for comparison.

		This Work	Experiment	Theory
CeO_2_	a	5.48	5.41 [20]	5.48 [21] 5.43 [22]
h-Ce_2_O_3_	a	3.89	3.89 [23]	3.94 [24] 3.92 [25]
	c	6.18	6.07 [23]	6.19 [24] 6.18 [25]
CeO_2_/Ce_2_O_3_	a	3.826		3.822 [14]
	c	15.546		15.351 [14]

**Table 2 materials-12-04041-t002:** Oxygen vacancy formation energies for various oxygen locations in the mixed valence state model and around a Ce atom in CeO_2_.

CeO_2_/Ce_2_O_3_	$V_{O1}$	$V_{O2}$	$V_{O3}$	$V_{O4}$	$V_{O5}$
E_f_ (eV)	4.79	4.91	5.25	5.27	5.32
CeO_2_	$V_{O}^{1st}$	$V_{O}^{2nd}$	$V_{O}^{3rd}$	$V_{O}^{4th}$	
E_f_ (eV)	4.49	5.17	5.57	5.82	

**Table 3 materials-12-04041-t003:** Bader’s total charge analysis for CeO_2_, Ce_2_O_3_, and the mixed valence structure.

System	Atom	Bader Charge
**CeO_2_**	O	7.19
	Ce	9.60
**CeO_2_-Ce_2_O_3_**	O1	7.21
	O2	7.22
	O3	7.22
	O4	7.31
	O5	7.40
	Ce1	9.56
	Ce2	9.58
	Ce3	9.97
	Ce4	9.99
	Ce5	9.63
**Ce_2_O_3_**	O1	7.39
	O2	7.31
	Ce	9.99

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
