# Peer review of "Influence of Mixed Valence on the Formation of Oxygen Vacancy in Cerium Oxides"

_materials, 2019, doi:10.3390/ma12244041_

Round 1
Reviewer 1 Report
In this manuscript, the authors suggest that the formation of oxygen vacancies can be an indicator of the charge status of cerium oxides CeO2, Ce2O3, and their intermixtures. Although the relation between the defect formation energy and the Ce charge status is not that rigorously suggested, I think the suggestion is interesting enough to be published in a peer-review journal like Materials, and the results are well presented with a decent writing. I would like to recommend the current manuscript for the publication in Materials, but there are some concerns about computational details employed in this work.
DFT+U method is used to capture strong electron correlation within Ce f-orbitals, but in this work the authors did not consider spin polarization. Figure 4 in the manuscript show no up/down spin information, and the gap remains closed even within the Ce f-bands in the Ce3+ configuration (f^1). Because the electron correlation is strong within the Ce f-orbital, the f^1 configuration should lead to the formation of local moments and the Mott-insulating phase (see, for example, https://iopscience.iop.org/article/10.1209/0295-5075/84/57009/meta). While the Mott phase happens even in the paramagnetic regime, in DFT+U it is not possible because of the band picture enforcing metallic bands when odd-number of electrons are occupying time-reversal symmetric bands. I suspect that it is the case for the results presented in this manuscript.
It is true that, rare-earth elements weakly hybridizes with surrounding oxygens, so the f-metallic states may not affect structural properties of ceria, but it needs to be checked whether the spurious metallic states affects the key results of this work. One reasonable way to mimic the Mott-physics is to allow spins to polarize (ISPIN=2 in INCAR), assume a magnetic order, and check whether the conclusion does remain unaffected. I guess that, because the magnetic exchange energy scales are order of few meV, so assuming any magnetic order should not affect formation energies which is order of ~ eV. Also, for the careful check, it is favorable for the spin-orbit coupling to be turned on (LSORBIT=.TRUE. in INCAR) due to the strong spin-orbit coupling in rare-earth elements.
And, in the second page there are a few sentences which is hard to understand; "When the Ce valence state changes from 4+ to 3+, such as in Ce2O3, it will become more difficult to form an oxygen vacancy around a Ce3+. On the other hand, if oxygen vacancies are continuously introduced around a 4+ Ce in CeO2, the formation of oxygen vacancy will become harder and harder. At a point, the formation energy of oxygen vacancy around a Ce4+ in CeO2 reaches the one around a Ce3+ in Ce2O3. Then we can consider the Ce has changed its valence state from 4+ to 3+. In this paper, we propose the formation energy of oxygen vacancy as a descriptor to determine the valence state of Ce in cerium oxides." Could the authors elaborate what this means in the revised manuscript?
Reviewer 2 Report
The manuscript "Signature of Mixed Valence in the Formation of Oxygen Vacancy in Cerium Oxides" contains some new information on the electronic structures of ceric oxides which might be useful in engineering materials with improved properties and so well fits into the journal. The body text itself seems to be coherent and well demonstrates the authors' results. Although it is well written the current form has some serious weaknesses and so a major revision is needed.
Around line 38-42: However, it is in the 'Introduction' section and that is not necessary the authors' opinion, the referee would suggest adding some critical comments. The claimed sentences are showing the ceric oxide(s) as dominantly covalent bound structure(s). The referee's opinion is different. According to some other theoretical papers (e.g. doi:10.1021/jp062886k and IUPAC list of elements' electronegativity) the EN(Ce0)=1.1, EN(Ce3+)=1.348, EN(Ce4+)=1.608, and EN(O)=3.4-3.5, EN(O2-)=3.6. These values are rather suggesting a strong ionic bond character than covalent ones. Please, insert a couple of sentences there to better understand not only the situation but a deeper rationale of the work.
The paper does not contain a 'Materials and Methods' section and so finding the (by the way: incomplete) details of the calculations are difficult.
The crucial point for the 'major revision' recommendation is that the authors used in their calculations the Vienna package. Although that is a sophisticated package but less common than other quantum chemical software. The authors also used solid-state calculations which are going to be a more and more important method in the simulation of the solid-state, particularly crystal structures, therefore the used keywords should be strictly presented. But, not only the used keywords are hidden, but the starting structure(s) is also missed to present, only the results are discussed, the comparison of the simulation results with the experimental crystal structures are also superficial. Please, proved the used input files (with the coordinates and simulation keywords) in a Supporting Information file. It is necessary to call the attention of the authors that the reproducibility by other researchers is an inevitable requirement for scientific publications. It is somehow understandable but not tolerable when concealing the details of computation, prevent the scientific community from verifying their results or extending their method to other materials.
Please, complete the manuscript with an SI containing all necessary information of reproducibility.
The referee has found some minor issues also in the main text.
Citation of ref 3. is strange, please, check it, while ref 22 contains a typo in the article title.
Reviewer 3 Report
The paper reports on an important subject but needs some changes.
For a start, the title should be modified. "Signature" is a strange word in the context. Perhaps "role" or "influence"?
The structure is also very strange. There is only one Section called "Introduction", no Experimental part, Results/Discussion or Conclusions. The article should be modified in order to include all those sections.
Reviewer 4 Report
This manuscript reports a DFT investigation of the oxidation state of Ce in CeO2 and Ce2O3 and how this can be used as an indicator of the local oxidation state of Ce in defective ceria.
The computational methods are adequate for the problem, the results are presented in a clear way and the discussion is physically sound.
In my experience, I can confirm that Bader charges of Ce4+ and Ce3+ differ only by 0.4e. The only minor point is on page 5, lines 130-131: it is not strictly true that "a common charge density contour" indicates a covalent bond between Ce and O. In reality one should perform a QTAIM analysis and the chracter of the Ce-O bond depends on the value of the laplacian of the density at the bond critical point.
In conclusion, I recommend publication of this paper.
Round 2
Reviewer 2 Report
The authors exhaustively answered the referee's concerns. Only one problem remained what the authors still left open: missing input files. Ok, the keyword problem has been answered but the starting geometries are still missing. They have missed informing the reader about their source. The voluntarily built structures a priori have subjectivity. If they used publicly available sophisticated sources why did not they reveal them for the reader? The authors mentioned that one of the referees confirmed their observation about the charge difference. Now, the referee feels also brave to share his own experiences: particularly the solid-state modeling of known crystalline structures is more reliable using real crystal structures than creating a structure by any software.
Please, provide the input or the source of structure files used in their calculations.
Author Response
We are grateful for this knowlegeable reviewer report on our manuscript. According to Reviewer's suggestion, we have provided the Supplementary Material in the resubmited version of our manuscript.